# Will Nothing Be the Same Again?: Changes in Lifestyle during COVID-19 Pandemic and Consequences on Mental Health

**DOI:** 10.3390/ijerph18168433

**Published:** 2021-08-10

**Authors:** Emanuele Caroppo, Marianna Mazza, Alessandra Sannella, Giuseppe Marano, Carla Avallone, Angelo Emilio Claro, Delfina Janiri, Lorenzo Moccia, Luigi Janiri, Gabriele Sani

**Affiliations:** 1Local Health Unit ROMA 2, Mental Health Department, 00159 Rome, Italy; emanuelecaroppo@gmail.com; 2Department of Neurosciences, Fondazione Policlinico Universitario A. Gemelli, IRCCS, 00168 Rome, Italy; giuseppemaranogm@gmail.com (G.M.); avallonecarla@yahoo.it (C.A.); dott.claro@gmail.com (A.E.C.); delfina.janiri@gmail.com (D.J.); lorenzomoccia27@gmail.com (L.M.); luigi.janiri@unicatt.it (L.J.); gabriele.sani@unicatt.it (G.S.); 3Psychiatry Unit, Università Cattolica del Sacro Cuore, 00168 Rome, Italy; 4Department of Human Sciences, Social and Health, University of Cassino and South Lazio, 03043 Cassino, Italy; alessandra.sannella@unicas.it

**Keywords:** COVID-19, lifestyle, unhealthy habits, mental health, personalized medicine

## Abstract

Social isolation caused by the COVID-19 pandemic has drastically affected lifestyles: from sedentary behaviors to reduced physical activity, from disrupted sleep patterns to altered dietary habits. As a consequence, serious mental and emotional responses have been registered. There was a significant decline in physical and other meaningful activities of daily living, leisure, social activity, and education. In children, collateral effects of the pandemic include inadequate nutrition with a risk of both overweight and underweight, addiction to screens, lack of schooling, and psychosocial difficulties. Older adults are frequently unable to adapt to lockdown measures and suffer from depression and cognitive complaints. Recent studies focusing on changes in lifestyle during the Covid-19 pandemic and consequences on mental health have been identified in PubMed/Medline, Scopus, Embase, and ScienceDirect. All the available literature has been retrospectively reviewed. The results of the present narrative review suggest that mental distress caused by social isolation seems to be linked not only to personality characteristics but also to several lifestyle components (sleep disruption, altered eating habits, reduced physical activity). This review aims to explore major changes in the lifestyle and quality of life and the impact of these changes on mental health, and to inform clinicians and policymakers about elements that may reduce the negative psychological effects of the quarantine period imposed during this worldwide crisis. There is an urgent need for tailored preventive, diagnostic, and therapeutic mental health interventions for the general population and for higher risk groups.

## 1. Introduction

The spread of the COVID-19 (SARS-CoV-2) pandemic all over the world has forced countries to handle the crisis in different ways, declaring a national state of alarm and establishing a mandatory home lockdown. The COVID-19 infection represents a strong stress stimulus, which has the capacity to induce high levels of perceived risk, fear, and anger, while forced quarantine at home may provoke an experience of boredom and loneliness, eliciting negative mental and behavioral responses in people [1]. It seems that the more time people remained at home, the more intense the resulting mental, emotional and lifestyle problems [2]. This situation has disrupted life and consequently altered multifaceted lifestyle behaviors. As a consequence, collateral damages of the pandemic are represented by inadequate nutrition with a risk of both overweight or underweight, addiction to screens, social isolation, disrupted sleep, and reduced physical activity with increased sedentariness: all these indirect effects of the COVID-19 outbreak have a potential mental health impact, particularly for vulnerable groups, and require effective and targeted measures. 

## 2. Methods

Studies focusing on changes in lifestyles during the COVID-19 pandemic and consequences on mental health were identified in PubMed/Medline, Scopus, Embase, and ScienceDirect. We searched cited databases for peer-reviewed publications related to the following keywords: “physical activity”, “sedentary behavior”, “sleep”, “lifestyle behaviors”, “COVID-19”, “lockdown”, “eating behavior”, “mental health” on 1st July 2021. Inclusion criteria included original studies in peer reviewed journals focusing on changes in lifestyle behaviors during the COVID-19 pandemic and consequent lockdown. Eligible studies had to report data on habit changes during the pandemic and mental health. Both longitudinal and cross-sectional studies were admitted and could be retrospective or prospective. There were no time limits or language limits in regard to the selection of appropriate studies. Reviews or meta-analyses focusing on changes in lifestyle and possible preventive strategies have also been consulted. The latter were used to search among their references for further possible eligible studies. Studies were excluded if they did not focus on or were unrelated to the subject matter and if they were case reports or series. All authors participated in the selection of eligible studies to include in the present review. 

## 3. Weight-Related Lifestyle Behaviors and the COVID-19 

During the worldwide COVID-19 crisis and lockdown restrictions, behaviors that are health-protective against weight gain such as eating a healthy diet may be more difficult to achieve and maintain. A decrease in dietary diversification, with an aggravating effect of lockdown on disrupted consumption patterns, elevated symptoms of generalized anxiety disorder, decreased physical activity levels, and perceived weight gain have the effect of enhancing the risk of overweight and obesity [3]. More time at home may cause additional eating, along with sedentariness. Stress related to fear and the continuous bombardment of news by the media about the spread of the pandemic may push one to consume so-called “comfort foods” (mainly composed of sugar or fats) or bring about a greater consumption of snacks between meals, with a consequent heightened risk of developing obesity [4]. In a cross-sectional survey conducted in the United Kingdom, 79% of participants reported a decline of at least one of five weight gain protective lifestyle behaviors studied (eating healthy, bingeing on food, exercising, sleep, alcohol consumption). In particular, subjects with a diagnosis of psychiatric illness or obesity resulted in an increased risk of weight gain during the COVID-19 crisis [5]. A web-based survey conducted in France suggests that weight gain may also be interpreted as the result of the observed increase in addiction-related habits (caloric/salty food intake, screen use, substance use) during lockdown [6]. Similarly, a Spanish study observed a rise in emotional eating during the months of confinement, “food craving” (the desire to consume a specific kind of food), and eating to compensate for boredom or anxiety with an increase in weight [7]. People living with obesity and mental health problems may have an increased risk of showing lifestyle behaviors associated with weight gain during the COVID-19 crisis [5]. It has been outlined that during lockdown, a higher BMI (body mass index) was predictive of greater overeating and lower physical activity [8]. 

## 4. Reduced Physical Activity during Lockdown 

Among abrupt societal changes related to the impact of COVID-19, there is a reduction of physical activity and prolonged sedentary behavior. Restrictions in physical activity were due to closed sport centers and limited social mobility. Social distancing and teleworking may contribute to sedentary lifestyles and an augmented sitting time during the day, together with less time spent engaging in leisurely vigorous physical activity and total physical activity, unfavorable changes in motivation, and individual perceptions of fatigue [9]. It is well known that regular physical activity helps prevent several chronic medical conditions, such as diabetes, hypertension, cardiovascular diseases, cancer, chronic kidney diseases, obesity, and osteoarthritis. The other side of the coin is that a reduced physical activity leads to an increased body weight and risk of illness, including inflammatory and cardiometabolic diseases, with a consequent higher risk of contracting infectious diseases. Regular physical activity has also been demonstrated to potentially protect mental health and increase quality of life [10]. Maintaining and enhancing physical activity participation may mitigate depressive and anxiety symptoms associated with self-isolation/quarantine [11], because mental health and physical health are strongly associated and affected by each other. From such perspective, it has been noted that pregnant women who reported exercise changes during the pandemic exhibited significantly higher prenatal depression scores compared to those reporting no changes [12]. It seems that resilience may buffer the deleterious impact of quarantine on physical activity [13] and that exercise may lead to healthier nutritional choices (fruit, vegetables, fish) and mediate the effects of a deflected mood on unhealthy dietary habits [14]. In fact, there is evidence that older adults who regularly engaged in physical activity during the quarantine reported higher scores in resilience and positive affect and a lower incidence of depressive symptoms [15]. Besides, since there is an important correlation between sedentary behavior and low mood, the issue of reducing sedentary time during the period of social isolation may be stressed for beneficial effects during lockdown but also for future wellbeing [16]. People should be encouraged to maintain a moderate amount of physical activity during the quarantine, particularly women, who are usually less active than men [17]. Since COVID-19 women were less physically active than men and reported more barriers and fewer facilitators to physical activity, they experienced significantly more generalized anxiety than men and showed significantly lower mental health scores [18]. It has been observed that the restoration of physical activity through short-term interventions is not sufficient to improve mental health, so longer interventions are needed [19]. Since it has been demonstrated that maintaining or introducing an adequate level of physical activity is likely to mitigate detrimental effects of mental and physical problems related to the COVID-19 pandemic, promoting safe practice of physical activity in this difficult moment should represent a public health priority to promote better mental health and well-being [20,21]. 

## 5. Sleep Disruptions Due to COVID-19

Sleep disturbances have affected a great amount of people around the world during the COVID-19 pandemic lockdown. The loss of daily routines due to home confinement and the presence of change in work, family habits and financial concerns, the limited exposure to natural light, and reduced opportunities to exercise may have negative effects on sleep. Alterations in daily schedules have impacted circadian rhythms and energy balance with a significant repercussion of confinement on several external synchronizers of the biological clock [22]. More frequently observed sleep symptoms have been insomnia/disrupted sleep, daytime symptoms such as dozing off unintentionally in the day, difficulties falling/staying asleep, later bedtimes, abnormal behaviors in sleep, sleep-disordered breathing, restless legs, sleep phase disturbances, and nightmares [23]. Not only sleep quantity but also sleep quality was found to be compromised during the pandemic [24]. An Italian study found that more than half of the population had an impaired sleep quality and sleep habits during the COVID-19 lockdown; related risk factors for poor sleepers were female gender, living in Central Italy, loss of a close one because of COVID-19 infection, changed sleep-wake rhythms, elevated levels of stress, anxiety, and depression [25]. A study conducted in South Korea has demonstrated that the total time participants spent sleeping was significantly higher than that before the pandemic; nevertheless, since satisfaction with sleep decreased, they may have had a poor sleep quality [26]. Particularly, in students the increased use of social media applications led to a significant delay in falling asleep, usually at much later hours than usual, a lengthening of the duration of sleep and a general feeling of tiredness [27]. A reported impact on mental health (depressive symptoms and anxiety) was most strongly associated with more difficulties falling asleep, sleep disruption, nightmares, and daytime sleepiness. It has been suggested that worsening sleep quality may partly mediate the association between sedentary behaviors (physical inactivity, high TV viewing, high computer/tablet use) and mental health indicators (loneliness, sadness, anxiety) [28]. 

## 6. Consequences of COVID-19 Lockdown on Lifestyle Behaviors of Children and Adolescents 

The closure of schools due to lockdown has reduced possibilities for physical activities and social life. Children and adolescents have been deprived for a long time of educational environments, social activities, and consequently contact with peers, with a disruption of daily schedules and a significant reduction of affective, cognitive, and physical stimuli. Decreased organized physical activity, increase in sedentariness, screen time, and consumption of caloric and sugary food with a consequent higher susceptibility to weight gain may enhance the great problem of childhood obesity [29].

This situation represents a risk for the mental health of schoolchildren.

In preschoolers, one has observed during quarantine a reduction of sleep efficiency, an increase in internalizing (i.e., antisocial behaviors) or externalizing problems (i.e., anxious or depressed behaviors), and a reduction of the total physical activity [30], while it has been demonstrated that higher levels of physical activity were associated with an improvement of the mood state among children and adolescents in the pandemic [31]. There is also concern about the finding that long periods of free-movement restrictions may negatively affect cardiorespiratory fitness in children and adolescents, a critical hallmark of health in youth, measured through a delay during COVID-19 confinement of the normal development of VO_2_ max (maximal oxygen uptake). High levels of VO_2_ max in childhood and adolescence are associated with lower values of cardiovascular risk factors (waist circumference, blood pressure, total cholesterol, body mass index) and lower odds of metabolic syndrome in later life; therefore, it is essential for youths to achieve sufficient levels of physical activity to preserve reliable health indicators [32]. 

In a cross-sectional study investigating the prevalence of lifestyle habits and mental health problems in Chinese adolescents during the COVID-19 pandemic, it has been observed that better nutritional patterns and moderate physical activity were both associated with lower levels of depressive and anxiety symptoms, while highly active physical activity was associated with lower levels of insomnia, depressive, and anxiety symptoms [33]. 

Studies reporting the indirect effects of the COVID-19 pandemic on nutrition in children outline an increased consumption of junk food, snacking, and sweets, and a decreased consumption of fresh foods and eating in response to boredom or anxiety. There are concerns about the imminent risk of increased pediatric obesity in middle- and high-income countries, while undernutrition is expected to deepen in poor countries already affected by a humanitarian crisis [34]. During the COVID-19 pandemic, alerts have been issued regarding global food insecurity, described as concern about access to adequate and sufficient amounts of affordable and nutritious food. During the pandemic, a tendency has been observed to buy packaged and long-lasting foods rather than fresh foods. Among children and adolescents, food insecurity greatly impacts nutritional habits, often predisposing one to eating disorders or exacerbating eating pathology (binge eating disorder, bulimia nervosa, secretive eating, night-time eating) [35]. 

Sedentary behavior may have serious consequences on existing and emerging psychopathology in children and adolescents, as it has been counted among possible risk factors for the development of insomnia, depression, anxiety, and psychosis [36]. The potential mental health benefits of maintaining a positive affect, engaging in physical activity and limiting leisure screen time have been highlighted for children during the pandemic, particularly for children with overweight/obesity [37]. There are some promising strategies to combat sedentary behavior in youths, for example the organization of public space options with individual physical distancing, exercise activities via live video conference calls, active-play video games that allow one to engage in indoor exercise activities, and above all adequate education for parents about the mental health benefits of regular activities [38]. A better understanding of students’ behavioral and socializing changes during COVID-19 lockdown results in being pivotal to programing critical and effective strategies for managing children’s mental health. Sleep and eating patterns, screen time, physical activity, and leisure seem to represent the most significant variables, influencing the many consequences of school closure and lockdown [39]. 

## 7. Discussion

A great amount of research agrees in affirming that the COVID-19 pandemic has had a negative impact on healthy and active lifestyles, with a contemporary and consequent decrease of mental health and quality of life. 

It seems that individuals who have been involved in more physical activity and have adopted healthy lifestyle dietary and sleep rules had a better mental health and reached a better physical health status. Gender represents a predictor of mental health, since females seem to mentally adapt worse to confinement resulting from the COVID-19 pandemic [40]. Besides, prepandemic primary systems and specific emotion regulation may act as a protective or risk factor for mental and physical well-being during and after lockdown: specifically, pre-existing stable depressive symptoms may negatively influence the possibility of adopting healthier and more adaptive behaviors [41].

Unhealthy lifestyle behaviors observed in the pandemic period are inevitably related to the potential development of chronic diseases, but these behaviors also closely interact with the mental health of individuals. For example, physical limitation and ineffective weight management are frequently associated with stress, anxiety, and depression [42]. Some authors have suggested that dramatic changes in physical activity, sleep, eating behaviors, time use, and mental health have no precedents and that worldwide the COVID-19 crisis has inevitably tightened the link between lifestyle behaviors and depression [19]. Psychological and social symptoms elicited by lockdown and fear of contagion strongly condition the normal function of subjects and may significantly deteriorate daily life activities. In adolescence in particular, a critical period of life characterized by profound physiological developmental modifications that lead to adulthood, the adherence to healthy lifestyle habits is of the upmost importance to guarantee future health outcomes. 

## 8. Conclusions

All studies considered in this review agree in outline that short- and long-term strategic plans regarding the problem of changes in lifestyle during the COVID-19 pandemic and consequences of these changes on the mental health of individuals are warranted. Information and interventions for individuals, communities, and healthcare institutions aimed at maintaining the healthiest lifestyle under quarantine should be guaranteed in order to prevent chronic diseases and psychiatric problems not only during the pandemic but also after the end of the outbreak, paying particular attention to children, adolescents, and at-risk groups (individuals with mental and/or physical health problems that existed before the spread of the COVID-19 pandemic, women, and older adults). For example, social media may play an important role in facilitating the self-management of behaviors related to physical activity, diet, and quality of life [43]. Besides, the rapid implementation of large-scale urban transformations may increase access to public open spaces and active transport infrastructure with the aim to promote physical activity and reduce sedentariness [44]. It has been suggested that there are different patterns of lifestyle changes for people all over the world during the COVID-19 pandemic, so there is a need to tailor support, interventions, and advice to different population groups [45], with the aim of providing appropriate strategies to rebuild balanced lifestyle patterns. As an indicator in the study, it also wants to highlight the importance of physical activity in promoting health and mental health, following Goal 3 of the WHO 2030 agenda: health and well-being for all, and for all ages [46]. 

## Data Availability

The data presented in this study are available on request from the corresponding author.

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
