# Peer review of "Will Nothing Be the Same Again?: Changes in Lifestyle during COVID-19 Pandemic and Consequences on Mental Health"

_ijerph, 2021, doi:10.3390/ijerph18168433_

Round 1

Reviewer 1 Report

The COVID-19 pandemic and all subsequent government’s self-containment measures drastically affected individuals’ lifestyles (e.g., disrupted sleep patterns, unhealthy eating, and social isolation). This review aims to presents some of the major changes in lifestyle and quality of life and their impact on mental health to inform policymakers in order to reduce such negative psychological effects in the event of future quarantine periods during the COVID-19 crisis.

The review is well written and organized and suggests, based on the results collected, the need to design interventions for individuals in order to maintain a healthier lifestyle under quarantine to prevent chronic diseases and psychiatric problems, especially for children and adolescents and for at-risk groups.

I have only some minor suggestions to make:

  1. Line 68: the authors suggest that there was an “increases in food prices” all over the world however the piece of literature that is being cited reports such a trend only for Zimbabwe. I advise them to either erase such part or to report that it has been mainly observed in developing countries.
  2. Lines 71-75: This sentence is very long and I suggest dividing it to make it more readable-
  3. Line 93: was -> were
  4. Line 103: It is better to split the sentence in two “… [10]. Maintaining …”
  5. Line 114: instead of “low mood” I would suggest finding better wording.
  6. Line 138: ad -> had
  7. Line 163: during quarantine -> ,during quarantine,
  8. Line 233: menta -> mental

Author Response

Dear Editor,

We have revised our paper carefully following reviewers’ useful suggestions. Revised parts are in red in the text. In the present form we hope that our paper could be accepted for publication.

Reviewer #1:

The COVID-19 pandemic and all subsequent government’s self-containment measures drastically affected individuals’ lifestyles (e.g., disrupted sleep patterns, unhealthy eating, and social isolation). This review aims to presents some of the major changes in lifestyle and quality of life and their impact on mental health to inform policymakers in order to reduce such negative psychological effects in the event of future quarantine periods during the COVID-19 crisis. The review is well written and organized and suggests, based on the results collected, the need to design interventions for individuals in order to maintain a healthier lifestyle under quarantine to prevent chronic diseases and psychiatric problems, especially for children and adolescents and for at-risk groups.

Thank you for your appreciation.

I have only some minor suggestions to make:

Line 68: the authors suggest that there was an “increases in food prices” all over the world however the piece of literature that is being cited reports such a trend only for Zimbabwe. I advise them to either erase such part or to report that it has been mainly observed in developing countries.

“increases in food prices” has been deleted

Lines 71-75: This sentence is very long and I suggest dividing it to make it more readable-

Line 93: was -> were

Line 103: It is better to split the sentence in two “… [10]. Maintaining …”

Line 114: instead of “low mood” I would suggest finding better wording.

Line 138: ad -> had

Line 163: during quarantine -> ,during quarantine,

Line 233: menta -> mental

All suggested corrections have been made.

Reviewer 2 Report

This review describes major changes in the lifestyle and quality of life due to COVID-19 pandemic and the impact of these changes on mental health. This represents an important topic. I’ve found the paper up-to-date and clear.

For completeness, I suggest to cite and discuss some recent interesting papers:

For sleep disruptions:

Anti-COVID-19 measures threaten our healthy body weight: Changes in sleep and external synchronizers of circadian clocks during confinement. Baquerizo-Sedano L, Chaquila JA, Aguilar L, Ordovás JM, González-Muniesa P, Garaulet M. Clin Nutr. 2021 Jun 25:S0261-5614(21)00315-0. doi: 10.1016/j.clnu.2021.06.019. Online ahead of print. PMID: 34246488

For possible future solutions:

Social media use informing behaviours related to physical activity, diet and quality of life during COVID-19: a mixed methods study. Goodyear VA, Boardley I, Chiou SY, Fenton SAM, Makopoulou K, Stathi A, Wallis GA, Veldhuijzen van Zanten JJCS, Thompson JL. BMC Public Health. 2021 Jul 6;21(1):1333. doi: 10.1186/s12889-021-11398-0. PMID: 34229651

Scaling up urban infrastructure for physical activity in the COVID-19 pandemic and beyond. Jáuregui A, Lambert EV, Panter J, Moore C, Salvo D. Lancet. 2021 Jul 21:S0140-6736(21)01599-3. doi: 10.1016/S0140-6736(21)01599-3. Online ahead of print. PMID: 34302763

Discussion

In the discussions I would add a comment relating to the pre-pandemic affective dimension. In fact, studies report that pre-existing stable depressive systems influence the adoption of healthier and more adaptive behaviors.

Mariani, R., Renzi, A., Di Monte, C., Petrovska, E., & Di Trani, M. (2021). The Impact of the COVID-19 Pandemic on Primary Emotional Systems and Emotional Regulation. International journal of environmental research and public health18(11), 5742. https://doi.org/10.3390/ijerph18115742

There are English misspelling I would suggest a general English revision for easier reading

Author Response

Reviewer #2:

This review describes major changes in the lifestyle and quality of life due to COVID-19 pandemic and the impact of these changes on mental health. This represents an important topic. I’ve found the paper up-to-date and clear.

Thank you for your appreciation.

For completeness, I suggest to cite and discuss some recent interesting papers:

For sleep disruptions:

Anti-COVID-19 measures threaten our healthy body weight: Changes in sleep and external synchronizers of circadian clocks during confinement. Baquerizo-Sedano L, Chaquila JA, Aguilar L, Ordovás JM, González-Muniesa P, Garaulet M. Clin Nutr. 2021 Jun 25:S0261-5614(21)00315-0. doi: 10.1016/j.clnu.2021.06.019. Online ahead of print. PMID: 34246488

For possible future solutions:

Social media use informing behaviours related to physical activity, diet and quality of life during COVID-19: a mixed methods study. Goodyear VA, Boardley I, Chiou SY, Fenton SAM, Makopoulou K, Stathi A, Wallis GA, Veldhuijzen van Zanten JJCS, Thompson JL. BMC Public Health. 2021 Jul 6;21(1):1333. doi: 10.1186/s12889-021-11398-0. PMID: 34229651

Scaling up urban infrastructure for physical activity in the COVID-19 pandemic and beyond. Jáuregui A, Lambert EV, Panter J, Moore C, Salvo D. Lancet. 2021 Jul 21:S0140-6736(21)01599-3. doi: 10.1016/S0140-6736(21)01599-3. Online ahead of print. PMID: 34302763

Suggested literature has been added and discussed.

Discussion

In the discussions I would add a comment relating to the pre-pandemic affective dimension. In fact, studies report that pre-existing stable depressive systems influence the adoption of healthier and more adaptive behaviors.

Mariani, R., Renzi, A., Di Monte, C., Petrovska, E., & Di Trani, M. (2021). The Impact of the COVID-19 Pandemic on Primary Emotional Systems and Emotional Regulation. International journal of environmental research and public health, 18(11), 5742. https://doi.org/10.3390/ijerph18115742

Thank you for this useful suggestion. It has been added in the discussion.

 There are English misspelling I would suggest a general English revision for easier reading.

All paper has been completely revised for English language.